# Nephrotic Syndrome: From Pathophysiology to Novel Therapeutic Approaches

**DOI:** 10.3390/biomedicines12030569

**Published:** 2024-03-03

**Authors:** Valentina-Georgiana Frățilă, Gabriela Lupușoru, Bogdan Marian Sorohan, Bogdan Obrișcă, Valentin Mocanu, Mircea Lupușoru, Gener Ismail

**Affiliations:** 1Department of Nephrology, Carol Davila University of Medicine and Pharmacy, 020021 Bucharest, Romania; valentina-georgiana.fratila@drd.umfcd.ro (V.-G.F.); bogdan.sorohan@umfcd.ro (B.M.S.); bogdan.obrisca@umfcd.ro (B.O.); valentin-dumitrel.mocanu@drd.umfcd.ro (V.M.); gener.ismail@umfcd.ro (G.I.); 2Department of Nephrology, Fundeni Clinical Institute, 022328 Bucharest, Romania; 3Department of Physiology, Carol Davila University of Medicine and Pharmacy, 020021 Bucharest, Romania; mircea.lupusoru@umfcd.ro

**Keywords:** nephrotic syndrome, underfill and overfill hypothesis, serin protease, diuretic resistance, ENaC-blockers

## Abstract

Nephrotic edema stands out as one of the most common complications of nephrotic syndrome. The effective management of hypervolemia is paramount in addressing this condition. Initially, “the underfill hypothesis” suggested that proteinuria and hypoalbuminemia led to fluid extravasation into the interstitial space, causing the intravascular hypovolemia and activation of neurohormonal compensatory mechanisms, which increased the retention of salt and water. Consequently, the recommended management involved diuretics and human-albumin infusion. However, recent findings from human and animal studies have unveiled a kidney-limited sodium-reabsorption mechanism, attributed to the presence of various serine proteases in the tubular lumen-activating ENaC channels, thereby causing sodium reabsorption. There is currently no standardized guideline for diuretic therapy. In clinical practice, loop diuretics continue to be the preferred initial choice. It is noteworthy that patients often exhibit diuretic resistance due to various factors such as high-sodium diets, poor drug compliance, changes in pharmacokinetics or pharmacodynamics, kidney dysfunction, decreased renal flow, nephron remodeling and proteasuria. Considering these challenges, combining diuretics may be a rational approach to overcoming diuretic resistance. Despite the limited data available on diuretic treatment in nephrotic syndrome complicated by hypervolemia, ENaC blockers emerge as a potential add-on treatment for nephrotic edema.

## 1. Introduction

Edema is one of the key features of nephrotic syndrome, along with hypoalbuminemia, proteinuria and dyslipidemia. Edema is the main reason why patients with nephrotic syndrome initially consult a doctor. Nephrotic syndrome can affect both children and adults. Over the years, the incidence of nephrotic syndrome in adults has not changed, with approximately 2.7–3 new cases per 100,000 people per year [1]. Hypervolemia can have serious implications, such as symptomatic or asymptomatic pulmonary congestion, cardiovascular risk, hypertension, risk of local or systemic infections, and anasarca, including nephrosarca and impaired mobility [2]. Considering all these complications, the management of nephrotic edema should be as important as the treatment of nephrotic syndrome. 

## 2. Edema Development: The Underfill and Overfill Hypotheses

Two main hypotheses have been proposed to explain the development of edema in individuals with nephrotic syndrome: the underfill hypothesis and the overfill hypothesis. These two hypotheses are not entirely separated but actually interact with each other, with one of them being more pronounced in individual patients. Determining whether edema is due to underfilling or overfilling is the most important step in the treatment of nephrotic edema because, ultimately, the goal of treatment is to eliminate excess water with minimal complications [3,4].

### 2.1. The Underfill Hypothesis

The basic mechanism of the underfill hypothesis is explained by proteinuria leading to hypoalbuminemia, which results in reduced plasma oncotic pressure, which further induces fluid to leak into the interstitial space and thus causes edema. The movement of fluid from the intravascular space leads to intravascular hypovolemia, which activates compensatory neurohormonal mechanisms such as the renin-angiotensin-aldosterone system (RAAS) and increased vasopressin with salt and water retention, thus exacerbating edema [5]. The migration of water from one compartment to another is linked to the close relationship between the hydrostatic and oncotic pressures of capillary and interstitial fluids, which can be explained by the presence of Starling forces [6,7]. There seem to be differences between these pressures in relation to age—some children tend to have lower oncotic capillary pressure than most adults, as is seen in the case of children with MCD [3] and the etiology of nephrotic syndrome, including (1) various permeability factors with different influences on protein permeability [8], (2) different plasma or interstitial oncotic pressures [9,10,11], and (3) an increase in capillary hydrostatic pressure secondary to sodium retention, as in acute poststreptococcal glomerulonephritis [3,12].

An accurate clinical assessment of volume status remains problematic due to heterogeneity in the volume overload [3]. In a study of 43 children who experienced nephrotic-syndrome relapse, edema development was found to be a dynamic process involving three steps: initial proteinuria without edema but with increased aldosterone levels, increased renal flow and sodium retention; hypovolemia with edema and persistently increased aldosterone levels and sodium retention; and, ultimately, edema without hypovolemia or sodium retention and normal plasma aldosterone [13]. At first, edema is absent because of compensatory factors such as increased lymphatic flow and increased interstitial hydrostatic pressure due to fluid movement from the intravascular space, and these factors simultaneously act like a protective shield against further fluid entry and reduced interstitial oncotic pressure, which thus minimizes the transcapillary pressure gradient [3,14,15,16].

#### 2.1.1. Hypoalbuminemia

As we have mentioned before, a decreased serum albumin concentration lowers capillary oncotic pressure, leading to hypovolemia. Normally, we expect the correction of hypoalbuminemia to help overcome water retention. In contrast, patients with congenital analbuminemia do not appear to present with salt or water retention with subsequent edema [17,18]. Moreover, in patients receiving steroid treatment, volume tends to correct itself well before albumin does [19].

Considering how furosemide binds to albumin to reach the proximal tubule, where it is secreted into the intratubular space, it is reasonable to expect potential benefits from albumin infusion [5]. Duffy et al. examined various randomized controlled trials (RCTs) and crossover studies, but the results were conflicting—they were either positive or negative, with little effect on volume status. The authors emphasized that all the studies involved broad patient selection; therefore, whether albumin is useful in patients with diuretic resistance was not established. Therefore, a clear definition of diuretic resistance is of utmost importance, and future studies should focus on this topic [20]. A recent Cochrane review also found only one small relevant RCT, from which it was not possible to determine the role of albumin compared to no treatment, other supportive care, or albumin in combination with diuretics [21].

The possible mechanisms postulated by the authors for the increased diuresis and natriuresis highlighted in various studies include increased renal plasma flow and/or the direct tubular effects of diuretics, especially furosemide, when used in combination with albumin. These benefits appear to occur shortly after the infusion of albumin, i.e., between 6 and 24 h. Possible explanations for the wide range of outcomes are the etiology of nephrotic syndrome, age, differences in hypervolemic status, serum albumin and serum creatinine levels, salt intake, and different treatment approaches [22,23,24,25,26]. However, whether albumin infusion is beneficial for treating nephrotic edema has not been determined.

#### 2.1.2. Hypovolemia

Hypovolemia due to hypoalbuminemia leads to salt and water retention through the RAAS and vasopressin, resulting in edema. Studies using radioactive albumin in patients with nephrotic syndrome showed normal or increased plasma volume, implying that hypovolemia is only a minor cause of salt retention; thus, other possible mechanisms may be involved in salt and water retention [27].

#### 2.1.3. Renin-Angiotensin-Aldosterone System

Studies have shown that patients with nephrotic syndrome can have high, low, or even normal renin levels, mainly because edema is a dynamic process. The RAAS is also influenced by treatment with diuretics and the etiology of the disease. Diabetes mellitus is a known disease that can lead to the suppression of the RAAS [28,29]. Given this background, it is difficult to construct a homogeneous study population from which to draw conclusions. An analysis of an untreated cohort with prompt onset of edema showed that proximal and distal sodium reabsorption, glomerular filtration capacity, and diffuse capillary leakage are important factors that act differently in patients with low versus high renin levels. The cause of nephrotic syndrome may be related to different factors. For example, patients with high renin levels have tubular dysfunction leading to sodium retention and a generalized increase in capillary leakage that prevents volume expansion, changes most commonly observed in MCD. Patients with low renin levels have decreased sodium excretion due to impaired glomerular filtration, leading to volume expansion, as is the case for most glomerulonephritides [29].

It is understandable that the inhibition of either component of the RAAS would correct a hypervolemic state.

To our knowledge, there are no large randomized trials of renin suppression with direct inhibitors or beta-blockers. Meltzer et al. showed no increase in diuresis or natriuresis after propranolol treatment [29];The role of angiotensin II in the proximal tubular uptake of sodium via angiotensin II receptor type 1 (AT1) in nephrotic syndrome is controversial. Instead, angiotensin II increases sodium reabsorption in the cortical-collecting duct (CCD) system through epithelial sodium channel (ENaC) aldosterone-independent stimulation [30]. Nevertheless, angiotensin-converting enzyme-inhibitors (ACE-inhibitors), particularly captopril, failed to increase sodium and water excretion, although marked diuresis was noted in healthy subjects [31];Aldosterone antagonists have been shown to cause little or no improvement in the natriuretic effect in nephrotic patients or murine models [32,33,34]. Shapiro et al. demonstrated that spironolactone caused significant increase in sodium excretion in nephrotic patients compared to a placebo [24]. However, it is essential to consider that this was a small-scale study involving only five patients with nephrotic syndrome [35]. Moreover, in rats with unilateral puromycin aminonucleoside-induced (PAN-induced) nephrotic syndrome, sodium retention occurred only in the affected kidney, suggesting a localized mechanism to explain edema rather than a systemic factor such as hyperaldosteronism [36]. Additionally, in rats with PAN-induced nephrotic syndrome, ENaC activity was correlated with increased aldosterone levels, while adrenalectomized rats or corticosteroid-clamped rats maintained their sodium and water retention independent of hyperaldosteronism [37,38].

#### 2.1.4. Arginine Vasopressin

Arginine vasopressin [AVP; or antidiuretic hormone (ADH)] is produced in the hypothalamus and stored in the posterior lobe of the pituitary gland. The main function of AVP is to regulate the free-water balance via vasopressin receptor 2 (V2R) in the CCDs. AVP acts on specific receptors—vasopressin V1a (V1A) and V1b (V1B) receptors and V2R. V1A receptors are located in vascular smooth muscle and cause vasoconstriction. V1B receptors are located in the anterior pituitary and cause the release of adrenocorticotropic hormone, and V2 receptors are located in the CCDs and cause the absorption of free water via aquaporins [39]. Increased serum osmolality or decreased arterial blood volume stimulates AVP secretion, as may be the case in nephrotic edema [40]. However, there are studies and case reports indicating apparently elevated ADH in “overfill” patients, suggesting a possible pathological increase in ADH in nephrotic syndrome patients [41,42]. A vasopressin-receptor antagonist is an agent that interferes with action at the vasopressin receptor and induces effective aquaresis, unlike all other diuretics, which increase natriuresis. While these drugs are approved for other edematous conditions (heart and liver failure), the evidence for nephrotic edema is limited, and there are no large randomized trials [41]. Meena et al. showed that in 10 pediatric patients with steroid-resistant nephrotic syndrome and severe resistant edema, the combination of intravenous furosemide and oral tolvaptan increased urine output while decreasing body weight at 48 h. Renal function was not impaired, but three patients had hypernatremia, with a serum sodium concentration > 145 mEq/L. Vasopressin-receptor antagonists are associated with many other side effects (liver failure, risk of gastrointestinal bleeding, acute kidney injury (AKI), thirst, and dyselectrolytemia) and are not recommended as a means of first-line therapy or for routine use [43,44].

#### 2.1.5. Sympathetic Nervous System (SNS)

Physiological responses to hypovolemia include increased SNS activity, which in turn leads to increased renin secretion, increased tubular sodium reabsorption, and decreased renal blood flow. All these adaptive changes maintain the hypervolemic state [45]. Studies in various animal models of decreased sympathetic activity (the surgical interruption of nerve impulses, nerve destruction by phenol, and sympathetic blockading by phenoxybenzamine) have shown decreased sodium and water reabsorption, even in the absence of changes in hemodynamics [46]. Renal denervation similarly improved in a rat model of nephrotic syndrome, in which no marked suppression of sympathetic activity was previously observed following volume expansion. This finding suggested that there is another mechanism underlying the increased sympathetic activity present in nephrotic syndrome in addition to hypovolemia, namely, the inhibitory effect of the cardiopulmonary baroreflex on the sympathetic nervous system. The study showed a decreased cardiopulmonary reflex in nephrotic syndrome, with an attenuated inhibition of renal sympathetic activity, possibly due to increased RAAS activity [47].

#### 2.1.6. Atrial Natriuretic Peptide (ANP)

Volume expansion causes atrial-wall stretching, and, in response, cardiac muscle cells in the heart’s atria produce ANP. ANP is a hormone whose main function is to decrease extracellular fluid intake by stimulating natriuresis in the collecting duct. Research on humans and animals demonstrated that nephrotic edema has specific resistance to ANP [48]. There are currently three possible explanations for this. One explanation is the rapid degradation of ANP via cyclic guanosine monophosphate (cGMP) as a result of plasmin loss, which activates phosphodiesterase (PDE). This theory was supported in a mouse model of nephrotic syndrome, where the natriuretic response to volume expansion was restored upon the use of an inhibitor (PDE) [49]. Another possible explanation is an increase in levels of a cyclophilin-like protein, which decreases sodium excretion. Nephrotic syndrome and higher ANP levels cause an upregulation of this protein [50,51]. The final theory concerns corin, a serine/threonine protease that cleaves the pro-form of ANP into active ANP. For a long period, it was believed that only atrial cardiomyocytes secreted corin. Subsequent animal studies revealed that both ANP and corin are present in the kidney [52]. Polzin et al. provided evidence of decreased corin and ANP levels in the nephrotic kidney using two rat models. Moreover, higher ENaC activity was linked to lower corin levels. All these changes point to the importance of corin and ANP in treating nephrotic edema [53].

### 2.2. The Overfill Hypothesis

#### 2.2.1. Serin Proteases

The strongest evidence for a kidney-limiting mechanism of sodium retention in the presence of proteinuria is represented by cases of nephrotic syndrome in patients with low or normal renin and aldosterone levels and unilateral mouse models of nephrotic syndrome (adrenalectomized or PAN-induced nephrotic edema). New information on how certain filtered serine proteases work on various channels of the distal renal tubule to cause sodium retention has been obtained through human studies, molecular knowledge of the sodium channel ENaC, and animal models of nephrotic syndrome. When discussing the overfill hypothesis, it is important to note that there are three main targets: NHE3, ENaC, and Na/K ATPase.

Animal models have demonstrated increased NHE3 activity in nephrotic syndrome or non-nephrotic proteinuria. The megalin receptor and the inactive form of NHE3, which is bound to megalin, could also be the reason for this increase. In the case of proteinuria, megalin attempts to reabsorb proteins as much as possible, releasing NHE3. In Xenopus oocytes (OKP cells), NHE3 is recycled back to the cell surface from the endosome when albumin is present, leading to increased sodium retention in the proximal tubule [54,55].

In 1980, a kallikrein-like protease induced transepithelial sodium transport in the toad bladder, a model of the mammalian aldosterone-sensitive distal nephron [56]. This was the first evidence of sodium handling by serine proteases. Later, Vallet et al. demonstrated reduced sodium reabsorption in a Xenopus kidney cell line expressing ENaC after exposure to aprotinin. Adding trypsin or chymotrypsin caused a two- to threefold increase in sodium reabsorption. The authors concluded that these proteases either directly or indirectly (via hormonal, paracrine, or trafficking pathways) regulate the ENaC because it was not possible to demonstrate direct cleavage of the ENaC extracellular loop [57].

Further research in this area demonstrated that the ENaC expressed in oocytes can reabsorb sodium in the presence of low concentrations of trypsin or chymotrypsin [58]. The ENaC undergoes proteolysis due to trypsin. It is not activated by G proteins or enhanced channel expression on the oocyte surface, as previously believed [59]. The ENaC has three types of subunits—alpha (α), beta (β), and gamma (γ). Each subunit possesses two transmembrane domains that form an extracellular loop with both carboxyl and amino termini in the cytoplasm. The understanding of the ENaC structure led to later studies describing the mechanisms of action of various serine proteases.

In 2003, early evidence that ENaC maturation involves the proteolytic processing of the alpha and gamma subunits was demonstrated [60]. Subsequently, it was observed that gradual activation of the ENaC requires cleavage at different sites in the extracellular domains of both the alpha and gamma subunits. Proteolytic cleavage plays an important role in regulating the activity of these channels by increasing their probability of opening. For example, channels lacking proteolytic activity have a low probability of opening. Furin, a trans-Golgi network protease, causes an intermediate probability of opening by cleaving the alpha subunit of the ENaC at two different sites and its gamma subunit at one site [61]. Finally, a high probability of opening requires dual gamma-subunit cleavage via a second protease, such as prostasin [62]. Even though furin and prostasin play significant roles, there is a vast list of additional proteases that have the capacity to cleave the gamma subunit (transmembrane protease serine 4 (TMPRSS4), matriptase, cathepsin B, elastase, kallikrein and plasmin) [63]. Studies of aldosterone-infused rats, hypertensive model mice with proteinuria, and humans that demonstrated higher levels of furin-cleaved ENaC in diuretic-treated subjects provide additional evidence of ENaC proteolytic cleavage [64,65,66].

#### 2.2.2. Plasmin

Plasmin is the dominant ENaC-activating protease [67]. Experimental models of nephrotic syndrome have shown that plasmin is produced from activated plasminogen by urokinase-type plasminogen activator (uPA). The first in vivo evidence that uPA-plasmin causes sodium retention was obtained from podocin knockout mice. When uPA antibodies were used, sodium retention was reduced through a decrease in the activation of plasminogen. The same effect was observed when amiloride was used, apart from its inhibitory effect on ENaC itself [68]. Although plasmin is the main contributor to sodium and water retention, Bohnert et al. demonstrated that uPA activity is not mandatory for sodium retention. By using uPA knockout mice, they showed that hypervolemia still developed, mainly through the action of other proteases. However, the authors also emphasized that uPA knockout mice may contain traces of plasmin, which is produced by other proteases, such as kallikrein [69].

The concept of volume expansion due to serine proteases encompasses not only nephrotic syndrome but also other proteinuric clinical conditions, such as preeclampsia and diabetic nephropathy, and raises the question of whether an ENaC blockade with amiloride or the inhibition of serine protease activity is truly effective at lowering sodium reabsorption and thus blood pressure [70].

Recently, amiloride-sensitive sodium reabsorption, independent of any ENaC subunit, was demonstrated using a truncated variant of acid-sensing ion channel 2b (ASIC2b) in association with ASIC2a in a corticosteroid-clamped PAN-induced nephrotic-syndrome rat model (CC-PAN rat model). The expression of this ASIC2b variant was detected only in rats with nephrotic syndrome and was due to albumin endocytosis and the activation of the ERK-signaling pathway in all cell types that form CCDs except type A intercalated cells. The expression of this channel is also dependent on aldosterone. It appears that in a PAN nephrotic rat model with hyperaldosteronism, sodium reabsorption occurs mainly through the ENaC, whereas sodium reabsorption at normal aldosterone levels is ASIC2-dependent. CC-PAN nephrotic ASIC2b-null rats also did not exhibit sodium retention [71].

Accurate assessment of the predominant mechanism involved in nephrotic edema is difficult. Tests for certain laboratory markers, such as serum aldosterone, vasopressin, ANP, norepinephrine, and even urinary sodium and potassium, are usually unavailable in hospitals. These last two markers could help clinicians evaluate the transtubular potassium gradient (TTKG) index and fractional excretion of sodium (FENa). The TTKG index is an indirect measure of serum aldosterone. It increases in hypovolemia patients but decreases to less than 60% in nephrotic-syndrome patients with primary sodium retention, along with FENa > 0.5% and a suppressed aldosterone level. In most cases, certain clinical features (signs and symptoms of hypovolemia and hypoperfusion) and accessible laboratory markers (serum albumin, creatinine, estimated glomerular filtration rate (eGFR), and hematocrit) help physicians differentiate between underfill and overfill nephrotic edema. This differentiation is a very important step, as the treatment differs depending on the predominant underlying mechanism. While, in patients with hypervolemia, diuretic treatment is of utmost importance, in patients with underfill hypervolemia, excessive diuretic administration without albumin infusion can have serious deleterious consequences (AKI, dyselectrolytemia, etc.) [3,72]. There is no exact definition for diuretic resistance, but it is generally defined as the failure of diuretics to achieve decongestion despite the use of the maximum recommended doses, as evidenced by a low urinary sodium concentration. To date, there are no precise values for the maximum dose of diuretics or for the optimal urinary sodium level. Knauf and Mutschler in 1997 showed that an FENa of less than 0.2% in a state of hypervolemia due to any cause is associated with a poor response to diuretics and could be used as a clear definition of diuretic-resistant edema. However, this finding has not been validated in large clinical trials [73].

There are multiple causes of diuretic resistance, and these causes can be evaluated from a pharmacokinetic or pharmacodynamic perspective. Pharmacokinetics refers to all the factors that influence a diuretic’s ability to reach its site of action, such as hypoalbuminemia, proteinuria, or intestinal-wall edema. Pharmacodynamics describes how the kidney responds to a diuretic, such as increased tubular sodium reabsorption in the presence of various proteases (plasmin, furin, and plasminogen, etc.) [74,75,76,77,78].

## 3. Diuretic-Resistant Hypervolemia

There is no exact definition for diuretic resistance, but it is generally defined as the failure of diuretics to achieve decongestion despite the use of the maximum recommended doses, as evidenced by a low urinary sodium concentration. To date, there are no precise values for the maximum dose of diuretics or for the optimal urinary sodium level. Knauf and Mutschler 1997 showed that an FENa of less than 0.2% in a state of hypervolemia due to any cause is associated with a poor response to diuretics and could be used as a clear definition of diuretic-resistant edema. However, this finding has not been validated in large clinical trials [73].

There are multiple causes of diuretic resistance. Firstly, it is imperative to ensure the accuracy of the diagnosis of nephrotic syndrome and to exclude other potential causes of peripheral edema, including lymphedema or venous edema. Secondly, assessing patient compliance with prescribed medications and adherence to a low-salt diet is essential. Subsequently, it is important to identify factors contributing to the decreased transport of diuretics to the renal tubule, such as a low dosage or infrequent dosing of diuretics, impaired absorption due to intestinal-wall edema, or drug administration with food and low serum albumin levels. Diuretics circulate in the bloodstream bound to albumin. Consequently, hypoalbuminemia affect their distribution and delivery to the kidneys. Additionally, potential causes of reduced renal secretion of diuretics, such as hypovolemia resulting in decreased renal blood flow, uremia, and decreased kidney mass, should be evaluated. Finally, it is crucial to identify factors that may impair renal responsiveness, including the level of proteinuria and serin-proteases in urine, causing the activation of the ENaC channel, nephron remodeling resulting in increased sodium and water reabsorption, the activation of compensatory neurohormonal mechanisms (such as the renin-angiotensin-aldosterone system, sympathetic nervous system, and antidiuretic hormone), and the use of nonsteroidal anti-inflammatory drugs (NSAIDs) [64,65,66,67,68]. 

## 4. Diuretic Treatment

Diuretics are fundamental for relieving volume overload, but to date, there are no guidelines for the diuretic treatment of nephrotic edema. Diuretics comprise several classes whose mechanisms of action, pharmacokinetics, indications, and adverse effects are decisive for the choice of treatment. Their mechanism of action is represented in Figure 1.

Sulfonamide-loop diuretics, thiazide diuretics, and carbonic anhydrase inhibitors (CA inhibitors);Potassium-sparing diuretic-ENaC antagonists and aldosterone antagonists;Vasopressin-receptor antagonists (vaptans);Osmotic diuretics.

### 4.1. Loop Diuretics

Due to their high efficacy (increased sodium excretion of 25% of the total filtered sodium) and safety profile, loop diuretics are usually the first choice of treatment for hypervolemia. Furosemide is the most commonly used loop diuretic. Although it has variable oral bioavailability and a short half-life, furosemide is much more available than other loop diuretics (such as torsemide). The most frequent side effects of furosemide are hypokalemia, hyperchloremic metabolic alkalosis, hyperuricemia, hypo- or hypernatremia, and an increased risk of lithogenesis due to hypercalciuria. High doses of furosemide or furosemide in combination with aminoglycosides can cause ototoxicity [44,79].

### 4.2. Thiazide Diuretics

The second most commonly used class of diuretics is thiazides or thiazide-like diuretics. These drugs are categorized as moderate-efficacy drugs because they inhibit the reabsorption of only 5–6% of luminal sodium in the distal convoluted tubule. The widely used clinical agents are hydrochlorothiazide and indapamide, which are thiazide-like diuretics. However, the preferred choice is metolazone, a long-acting thiazide-like diuretic. Thiazides are often used to treat high blood pressure in patients with or without chronic kidney disease (CKD), even in those with advanced stages of CKD, as seen with chlorthalidone. The side effects of thiazide diuretics are similar to those of loop diuretics, except there is a greater risk of hyponatremia, hypokalemia, and hypomagnesemia with thiazides. Compared to loop diuretics, thiazides decrease urinary calcium levels and thus have a protective effect against kidney stones [80,81]. These drugs increase the level of serum uric acid and can also increase the risk of tumor lysis syndrome in patients with malignancies [82].

### 4.3. Potassium-Sparing Diuretics

These diuretics can be classified as ENaC-blockers or mineralocorticoid-receptor antagonists, and they are weak diuretics that inhibit less than 2% of the total filtered sodium load. These drugs are mostly used for their potassium-sparing effect and in patients with primary or secondary hyperaldosteronism. Mineralocorticoid-receptor antagonists or aldosterone-antagonists (spironolactone and eplerenone) are mainly used for patients with resistant hypertension, heart failure, and/or CKD. These drugs are able to reduce proteinuria, inflammation, and fibrosis and have been associated with a reduction in cardiovascular events [44,81]. The latest drug, finerenone, a nonsteroidal mineralocorticoid-receptor antagonist, showed a significant cardiorenal protective effect in two clinical trials, FIDELIO-DKD and FIGARO-DKD. Finerenone was recently approved for patients with CKD and type 2 diabetes mellitus [83,84]. The main side effect of finerenone is hyperkalemia [44].

### 4.4. Carbonic Anhydrase (CA) Inhibitors

These diuretics reduce sodium and water reabsorption in the proximal tubule, but most of the sodium and water is subsequently reabsorbed in the distal tubule. Therefore, these agents are not effective diuretics. Due to their ability to inhibit acid secretion, these compounds can be used in patients with metabolic alkalosis secondary to other diuretics. CA inhibitors are mostly used as antiglaucoma drugs that suppress the formation of aqueous humors in the eyes. Hypokalemia and metabolic acidosis are common side effects. They can also cause hypercalciuria, which favors the formation of calcium renal stones [44,85]. Recent randomized clinical trials have shown the benefit of acetazolamide in hypervolemic states of acute heart failure and nephrotic syndrome, but further studies are needed [86,87].

### 4.5. Vasopressin Receptor Antagonists (Vaptans)

These compounds are also known as aquaretics, and they can increase free-water clearance. Tolvaptan, a selective oral AVP V2-receptor antagonist, and conivaptan, a nonselective V1a/V2-receptor antagonist that is available intravenously, are both approved by the Food and Drug Administration (FDA) for the treatment of hypervolemic hyponatremia in congestive heart failure and euvolemic hyponatremia in the syndrome of inappropriate antidiuretic hormone secretion (SIADH). The results of the EVEREST trial failed to demonstrate the superiority of tolvaptan in terms of long-term clinical endpoints, but patients did have a better volume status. The FDA strongly advises clinicians against the use of tolvaptan for longer than 30 days and for patients with cirrhosis, except for patients with end-stage liver disease awaiting liver transplantation [44,79]. These drugs may be potential options for treating nephrotic edema because of their mechanism of action. As we have already discussed, only small studies and case reports have shown the benefit of vasopressin-receptor antagonists. The exact benefit of these drugs could not be demonstrated, as no measurements of sodium intake, vasopressin levels, or urine concentration were made. Therefore, we cannot say whether the vaptans are the exact cause of the resolution of edema [41]. To summarize, AVP antagonists are not widely available and are associated with high costs [5,44].

### 4.6. Osmotic Diuretics

Osmotic diuretics have different effects than all other diuretics in terms of their mechanism of action. They are freely filtered in the glomerulus and do not act on a specific tubular channel. Rather, these diuretics remain in the tubule and increase tubular osmotic pressure, which inhibits water reabsorption and disrupts countercurrent exchange and the medullary concentration gradient. Osmotic diuretics also cause cellular dehydration with intravascular expansion, which is why they are mostly used to reduce intracranial pressure in cerebral edema (mannitol) and to increase free-water excretion in hyponatremia (urea) [44].

### 4.7. Sodium–Glucose Cotransporter 2 (SGLT2)-Inhibitors

These drugs are also called gliflozins or flozins. They are a class of oral antidiabetic drugs that act on the SGLT2 protein expressed in the early proximal tubules to reduce the reabsorption of filtered glucose and sodium and promote urinary glucose excretion; thus, the non-reabsorbed glucose induces an osmotic diuretic effect [88]. Originally developed only as hypoglycemic agents, there is evidence that SGLT2-inhibitors can induce mild diuresis [89,90,91,92,93]. Even so, they are not used as first-line diuretics, but recent studies have shown that they can help with decongestion [94,95,96,97,98,99]. Thus, FDA-approved indications for SGLT2-inhibitors include heart failure across the full spectrum of left ventricular ejection fractions [100]. SGLT2-inhibitors have other benefits, such as slowing the progression of diabetic and nondiabetic CKD, possibly through reduced glomerular hyperfiltration and other pleiotropic physiological benefits [101,102,103,104,105,106]. Another potential benefit is that SGLT2-inhibitors help to correct hypervolemia-associated hyponatremia [107,108]. The most frequently reported adverse events are genital mycotic infections and urinary tract infections. Other significant adverse reactions to SGLT2-inhibitors include lower limb amputation, diabetic ketoacidosis, euglycemic diabetic ketoacidosis, AKI, Fournier gangrene, and hyperkalemia, especially when SGLT2-inhibitors are combined with ACE-inhibitors or angiotensin-receptor blockers (ARBs) in patients with renal impairment (Sodium–Glucose Transport Protein 2 (SGLT2)-Inhibitors—inderbir padda) [77,109].

Loop and thiazide diuretics can cause hyponatremia, hypokalemia, hypomagnesemia, and hyperchloremic metabolic alkalosis [110]. Vaptans and SGLT2-inhibitors use might be beneficial in counteracting hyponatremia [111,112,113], while MRAs’ or ENaC-blockers’ use might be beneficial in counteracting hypokalemia and metabolic alkalosis associated with loop and thiazide diuretics [114]. CA-inhibitors can be used as add-on therapy in case of metabolic alkalosis [74]. This association is also valid for reverse effects, as illustrated in Figure 2.

In finding the correct dosage and timing of diuretics, it can take time for clinicians to titrate and adjust the dose based on their individual experience when treating hypervolemia of any cause, especially when patients have poor kidney function. There is no equation that helps in this matter. Therefore, treatment generally starts with lower doses of diuretics. When adequate diuresis does not occur, a stepped-care approach is recommended, with gradually increasing doses, based on certain clinical and biological aspects (urine output, hydration status, weight, blood pressure, electrolytes, and serum creatinine, etc.) [1]. As we have already mentioned, loop diuretics are the drugs of choice. The aim is to increase urine output over the next 2–4 h. Failure to do so means that the natriuretic threshold has not been reached. In current clinical practice, the initial dose is doubled and subsequently increased up to the maximum dose of the diuretic, or the route of administration is switched to intravenous administration [5]. If diuretic resistance is present, the use of a second diuretic drug that acts on a different nephron segment is often effective. Drugs from a different class may act synergistically with the first by blocking the adaptive processes that limit the efficacy of diuretics, such as the activation of the RAAS and SNS, excessive NaCl consumption and the remodeling of the distal nephron. Distal convoluted tubule hypertrophy and hyperplasia occur due to the workload induced via diuretics [72,77,110,115].

#### 4.7.1. Managing Volume Overload in Certain Conditions

##### Heart Failure

The DOSE trial is the most important study that evaluated diuretic treatment in patients with acute decompensated heart failure. The study compared furosemide administered intravenously every 12 h via boluses or continuous infusion to furosemide administered at either high doses (2.5 times the previous oral dose) or low doses (equivalent to the patient’s previous oral dose). There was no significant difference between the treatment groups in terms of the primary efficacy or safety endpoints. However, there was a nonsignificant trend toward greater improvement in the global assessment of symptoms in the high-dose group [116]. The limited evidence to guide diuretic therapy is reflected in practice guidelines. The ACCF/AHA 2022 guideline for the management of heart failure assigns diuretics a class-I recommendation, but it does so based on level-B evidence. A loop-diuretic dose should be attempted first up to a maximum daily dose of 600 mg of furosemide, and a thiazide may be added if congestion persists [117]. In their most recent review, Novak and Ellison recommended that loop diuretics be given intravenously with gradually increasing doses, similar to the procedure used in the CARRESS-HF study [44,118]. 

Furosemide has a short half-life (6 h) and must be administered twice daily [79]. In patients with advanced CKD, torsemide may be preferable to furosemide because torsemide has a longer duration of action and is a long-acting thiazide (metolazone) [2,3]. Due to the 6 h half-life of furosemide, sodium is reabsorbed into the bloodstream during this period [79]. The addition of a long-acting thiazide diuretic (with a half-life of 14 to 50 h) therefore reduces sodium retention. In patients with persistent hypervolemia, reduced urine output, or electrolyte imbalance (most frequently hypokalemia), a potassium-sparing diuretic can be added—either an ENaC-blocker or a mineralocorticoid-receptor antagonist [119,120].

Recent data suggest that SGLT2 inhibitors can be used as an add-on therapy [44]. Large randomized clinical trials have demonstrated the cardio- and reno-protective effects of different SGLT2-inhibitors in patients with type 2 diabetes and in patients with CKD of other etiologies [101,103,104,105,121,122,123,124,125]. Several human and animal studies have shown that these compounds can modestly increase urinary sodium excretion and urine volume [89,90,91,126,127,128]. As a result, heart failure is now a class-I indication for the use of SGLT2-inhibitors [100].

ADVOR, a recent randomized trial of 519 patients with acute decompensated heart failure and clinical signs of hypervolemia, showed that the addition of intravenous acetazolamide at a dose of 500 mg/day to a loop diuretic improved decongestion more rapidly and without additional side effects than the placebo did [86].

##### End-Stage Liver Disease

Patients with liver cirrhosis have a reduced effective blood pressure, which causes hyperaldosteronism. As a result, patients experience fluid overload [129]. Mineralocorticoid-receptor antagonist therapy with spironolactone can be used as a first-line treatment, as it directly antagonizes the increase in RAAS activity [130]. Spironolactone is generally used at doses up to 400 mg/d. The goal of this regimen is a weight reduction of approximately 1 l/d. In general, an aggressive approach is not recommended due to the high risk of intravascular hypovolemia and AKI if the patient is hemodynamically stable [131]. As with heart failure, if decongestion is inadequate following spironolactone, the next step is the administration of a second diuretic. Clinicians usually choose a thiazide diuretic, while a loop diuretic is chosen as a last resort. A loop diuretic is usually last in the list of diuretic options as it has a low diuretic effect. This is due to their strong activation of the RAAS rather than their insufficient tubular excretion (except in patients with CKD). Thus, in patients with liver cirrhosis, it is important to increase the frequency of loop diuretics because higher doses are not effective [74,119]. Standard doses of diuretics include spironolactone 100 mg/d and furosemide 40 mg/d [44].

##### Nephrotic Syndrome

The management of nephrotic edema continues to rely predominantly on clinical expertise, as evidence from large-scale trials is currently lacking. Consequently, the absence of comprehensive data precludes researchers from issuing definitive guidelines for effectively addressing fluid overload in nephrotic-syndrome patients. Loop diuretics remain the first-line therapy for nephrotic syndrome, with intravenous administration being the preferred route owing to limited oral absorption caused by intestinal-wall edema. The standard initial dose of oral furosemide is 2 mg/kg/d. In the absence of an increase in diuresis within the next 2–4 h, the dosage may be increased to 6 mg/kg/d. If there is still an inadequate response to the maximum oral dose despite these adjustments, switching to intravenous administration is recommended. Combination diuretic therapy using any of several thiazide-type diuretics can induce weight loss and edema resolution if furosemide alone is not sufficient to control fluid retention. Clinicians should properly monitor patients for any indication of hypovolemia, such as low blood pressure, high pulse rate, or increased hematocrit. In the event of hypovolemia, an albumin infusion should be considered for fluid resuscitation [5].

In 2017, Fallahzadeh et al. conducted a randomized trial involving 20 patients with refractory nephrotic edema in which the efficacy of preloading with acetazolamide and hydrochlorothiazide was compared to that of preloading with furosemide and hydrochlorothiazide over a one-week period. Following this preloading phase, patients in both treatment arms received 40 mg of furosemide for a two-week period. The authors demonstrated that the combination of acetazolamide and hydrochlorothiazide improved diuresis, as indicated by differences in the mean weight change and urinary volume. These authors emphasized the importance of pendrin in nephrotic syndrome, paralleling its importance to the ENaC. It is imperative to mention that a notable limitation of the study was the absence of serum bicarbonate and chloride measurements; therefore, there was a potential risk of the patients developing metabolic acidosis [87].

As previously noted, the urinary excretion of serine proteases in patients with nephrotic syndrome means that the ENaC can be activated in these patients [132]. Consequently, therapeutic interventions utilizing ENaC blockers, such as amiloride and triamterene, may yield a more favorable response [5]. Amiloride exhibits an additional benefit by decreasing plasmin levels through the inhibition of the urokinase plasminogen-activator receptor (uPAR) [133,134,135,136]. However, evidence from randomized studies is lacking. There are a limited number of case reports showing the therapeutic benefits of these drugs, even in patients with resistant hypervolemia [137,138]. Additionally, several clinical trials in both diabetic and nondiabetic populations have demonstrated improved blood pressure control or a significant or nonsignificant weight loss with the addition of amiloride [139,140,141,142,143,144,145,146]. Although hyperkalemia is the most frequently observed adverse event associated with amiloride [63,147], clinicians need not be concerned about this, particularly in the absence of other risk factors for elevated potassium levels (such as high doses of amiloride, the coadministration of ACE inhibitors, or severe renal dysfunction) [144,148]. 

A single randomized trial encompassing 22 patients with refractory nephrotic edema that demonstrated a significant difference in weight change from baseline within the cohort administered an oral combination of diuretics comprising amiloride, hydrochlorothiazide, and furosemide, in contrast to those receiving intravenous furosemide −3.33 kg (95% CI: −6.34 to −0.31), *p* = 0.03] [149]. We may soon see the results of a phase-III clinical trial that compared the efficacy of amiloride (5 mg/d) for reducing nephrotic edema against that of standard therapy with furosemide (40 mg/d). Diuretics were administered over a 14-day period in patients with nephrotic edema and an eGFR > 30 mL/min/1.73 m^2^ (ClinicalTrials.gov Identifier: NCT05079789). Experimental animal studies and a documented case study of Fabry disease have demonstrated that amiloride has the potential to reduce proteinuria through an alternative mechanism, specifically through its inhibitory effect on uPAR [68,134,150,151,152,153]. uPAR is also implicated in the activation of αvβ3 integrin or the vitronectin receptor, resulting in podocyte contraction and subsequent detachment from the glomerulus, leading to proteinuria [154]. There are theories that suggest soluble uPAR affects proximal tubular cells and induces fibrosis in an integrin-dependent manner [155]. The reduction in uPAR levels due to the action of amiloride translates into a decrease in the concentration of soluble uPAR (suPAR), the circulating version of uPAR. This, in turn, inhibits the activation of αvβ3 integrin, ultimately contributing to a decrease in proteinuria [134,136,156].

An additional murine study demonstrated that the overexpression of ENaC occurs after pretreatment with acetazolamide, a carbonic anhydrase inhibitor, in NCC knockout mice. Amiloride significantly increased urine output, as this overexpression of ENaC becomes necessary for increased sodium delivery after sodium reabsorption is blocked in the proximal and distal convoluted tubules. These findings suggest that further investigation into the combined use of these three agents (acetazolamide, amiloride and thiazide diuretic) is warranted [157]. As nephron remodeling is a major cause of diuretic resistance, it is reasonable to consider diuretic combinations to overcome it. Various classes of diuretics may be associated with loop diuretics in cases of nephrotic edema or heart failure, while an aldosterone antagonist is typically the first choice in cirrhosis, as illustrated in Figure 3.

#### 4.7.2. New Pharmacological Targets

As previously stated, limitations are encountered in the use of diuretics to address fluid overload due to various factors, including severe hypervolemia, dyselectrolytemia, and kidney dysfunction. In such cases, ultrafiltration remains the lone viable option [77,79,158]. Recent attention has been directed toward novel and promising pharmacological interventions. These include serine protease inhibitors; adenosine A1-receptor antagonists; urea-transporter inhibitors; ROMK-inhibitors; WNK-SPAK-inhibitors; natriuretic peptide-receptor agonists; pendrin-inhibitors; guanylyl-cyclase A-receptor activators; and inhibitors of relaxin, luteolin, and epicatechin. These emerging therapeutic options hold the potential for overcoming diuretic resistance. Importantly, while some of these interventions have been investigated in animal studies, their feasibility and efficacy in humans have yet to be established, and further exploration in future research endeavors is warranted [74,77,159,160].

## 5. Conclusions

At present, loop diuretics remain the primary choice for treating disorders with expanded extracellular volume, including nephrotic syndrome. Some examples of loop diuretics include furosemide, bumetanide, and torsemide (Figure 4). Alongside thiazide diuretics, they were developed from 1,2,4-benzothiadiazine derivatives in the late 1950s. Carbonic anhydrase inhibitors constituted the initial modern diuretic option following mercurials. The observation that the antibiotic sulphanilamide increased urine output prompted the development of carbonic anhydrase inhibitors, achieved by substituting a carboxy group for the aromatic amino group of sulphanilamide, thus producing carboxybenzenesulphonamide. Further structural modifications of sulphanilamide-like compounds led to the synthesis of acetazolamide in 1954 [110,160,161]. Subsequent structural modification resulted in the creation of 6-chloro-2H-1,2,4-benzothiadiazine-7-sulphonamide-1,1-dioxide (chlorothiazide), the first thiazide diuretic. Furosemide, a sulphonamide derivative chemically known as 4-chloro-N-furfuryl-5-sulphamoyl anthranilic acid, was the inaugural approved loop diuretic in 1966, followed by bumetanide in 1983, and the most recent one, torsemide, a sulphonylurea, in 1993. Chlorothiazid served as the prototype for the most commonly used thiazide diuretics in contemporary practice, like hydrochlorothiazide, while modifications of the thiadiazine nucleus resulted in thiazide-like diuretics, such as indapamide, chlortalidone, and metolazone [110,160]. However, it is crucial to acknowledge that nephrotic syndrome exhibits two key features: the activation of ENaC channels due to various serine proteases and the potential for diuretic treatment with ENaC-blockers. Amiloride (3,5-diamino-N-carba- mimidoyl-6-chloropyrazine-2-carboxamide), a synthetic pyrazinoylguanidine derivative, stands as the representative diuretic of its class, while triamterene, a pteridine, possesses a reduced potency and increased nephrotoxicity compared to amiloride [110,139,144,162]. These blockers appear to be advantageous in overcoming diuretic resistance when used in combination with loop diuretics and other agents. Another benefit is their capacity to counter hypokalemia associated with loop and thiazide diuretics. Based on the evidence presented in this review, there is a compelling need for further exploration of the potential benefits of ENaC blockade through large-scale clinical trials.

## Figures and Tables

**Figure 1 biomedicines-12-00569-f001:**
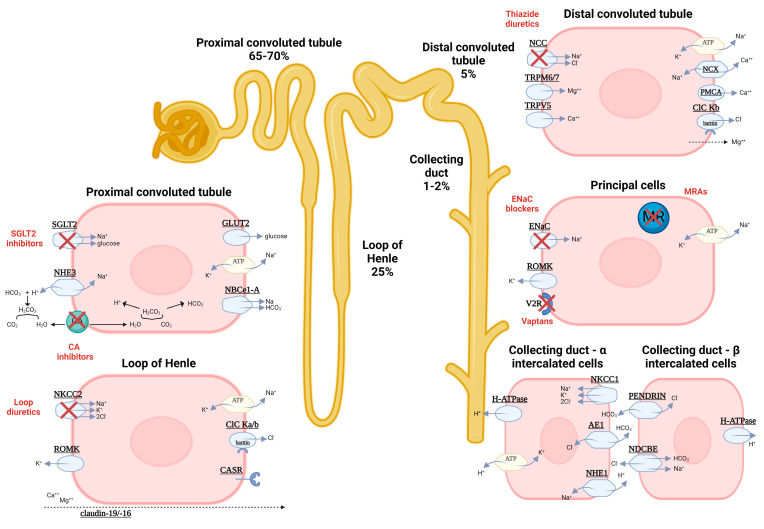
Classes of diuretics and their mechanism of action at different sites in the renal tubule; Abbreviations: carbonic anhydrase (CA), epithelial sodium-channel blockers (ENaCb), mineralocorticoid-receptor antagonists (MRAs), sodium–glucose transport protein 2 (SGLT2), classes of diuretics and their mechanism of action at different sites in the renal tubule; Abbreviations: carbonic anhydrase (CA); anion-exchanger 1 (AE1); aquaporin-2/4 (AQP-2/4); calcium-sensing receptor (CASR); chloride channels with barttin (ClC Ka/b); glucose-transporter 2 (GLUT2); epithelial sodium channels (ENaC); hydrogen ATPase (H+ ATPase); hydrogen-potassium ATPase (H+/K+ ATPase); mineralocorticoid receptor (MR); sodium-potassium pump (Na+/K+ ATPase); electrogenic sodium-bicarbonate cotransporter 1 (NBCe1-A); sodium-chloride symporter (NCC); Na-K-Cl cotransporter (NCCK2); sodium–calcium exchanger (NCX1); sodium-driven chloride/bicarbonate exchanger (NDCBE); sodium–hydrogen antiporter 1/3 (NHE1/3); Na-K-Cl cotransporter (NCCK1); Na(+)-independent Cl(-)/HCO3(-)-exchanger (pendrin); plasma-membrane Ca2+ ATPase (PMCA); renal outer medullary potassium channel (ROMK); sodium–glucose cotransporter-2 (SGLT2); transient receptor potential cation channel subfamily M member 6/7 (TRPM6/7); transient receptor potential cation channel subfamily V member 5 (TRPV5); vasopressin receptor 2 (V2R).

**Figure 2 biomedicines-12-00569-f002:**
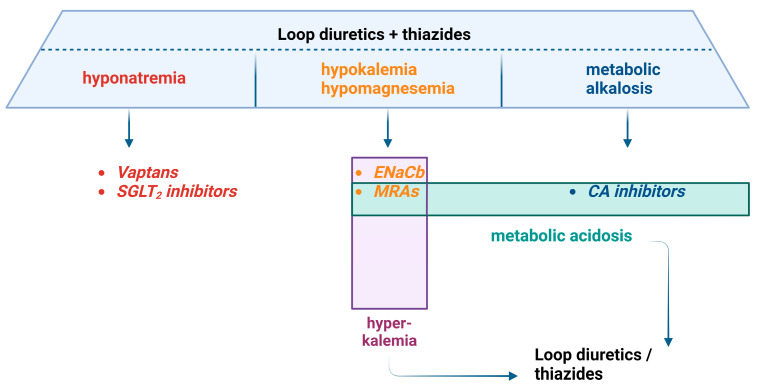
Major adverse events of diuretics and possible diuretic combinations to overcome them. Abbreviations: carbonic anhydrase (CA), epithelial sodium-channel blockers (ENaCb), mineralocorticoid-receptor antagonists (MRAs), sodium–glucose transport protein 2 (SGLT2).

**Figure 3 biomedicines-12-00569-f003:**
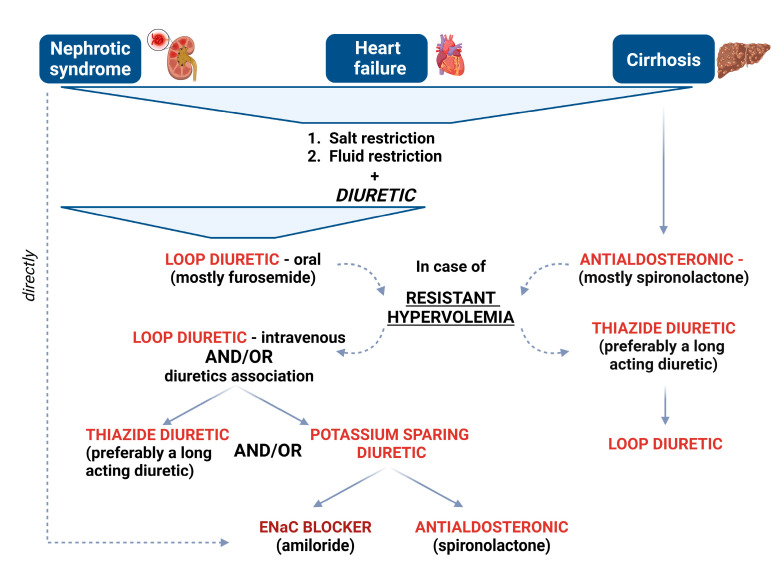
A stepwise approach for diuretic-resistance management of edematous states.

**Figure 4 biomedicines-12-00569-f004:**
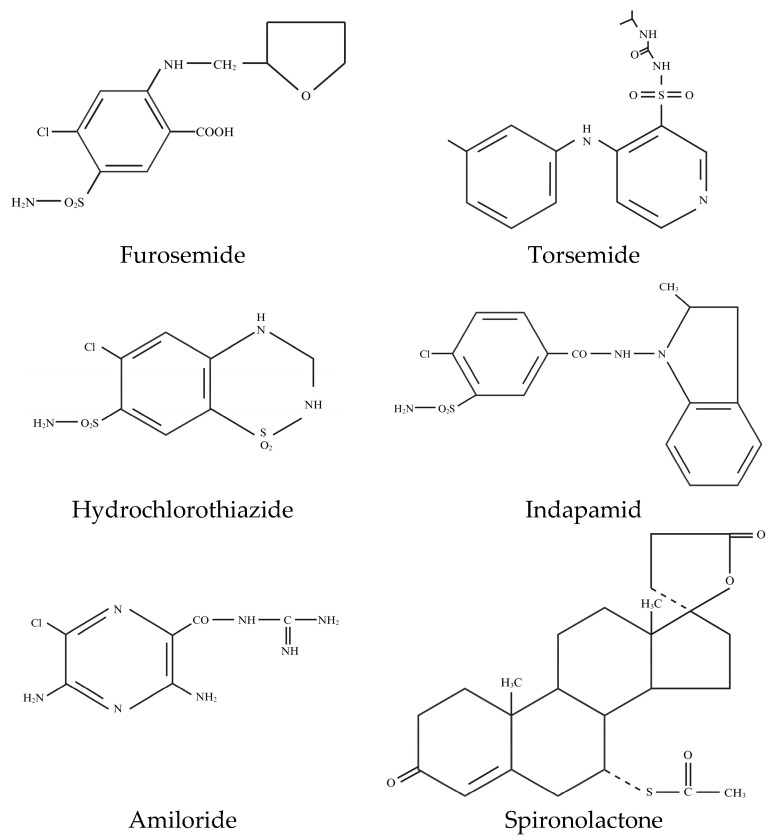
Structures of some loop diuretics, thizides or thiazide-like diuretics and potassium-sparing diuretics.

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
