# Peer review of "Nephrotic Syndrome: From Pathophysiology to Novel Therapeutic Approaches"

_biomedicines, 2024, doi:10.3390/biomedicines12030569_

Round 1

Reviewer 1 Report

Comments and Suggestions for Authors

Review for the manuscript “Nephrotic Syndrome: From Pathophysiology to Novel Therapeutic Approaches”.

An interesting example of review on nephrotic syndrome covering various sides of the disease and its treatment. Review includes hypotheses on the development of edema which is one of the main symptom of nephrotic syndrome and its treatment by diuretics.

There is one point should be addressed before this manuscript could be published in biomedicines journal. As soon as you conclude that “diuretics remain the primary choice for treating disorders … including nephrotic syndrome” structure formulas of the most notable diuretics should be added to the manuscript. This will made manuscript more descriptive, especially for chemists.

Author Response

Thank you once again for considering my submission. I am writing in response to your insightful suggestions regarding the structural formulas of prominent diuretics. We have duly revised a segment of the section “Conclusions” as per your guidance, and furthermore, integrated six structural formulas into a single figure (Figure 4). The revised text is denoted by red markings for clarity. If there are any further revisions or adjustments needed to facilitate its publication, please don't hesitate to let me know. I am more than willing to accommodate any suggestions or feedback.

Reviewer 2 Report

Comments and Suggestions for Authors

Intense research was performed recently to improve treatment of nephrotic syndrome. In the study of Fratila et al., the authors depict perspicuously the mechanisms responsible for formation of edema in nephrotic syndrome. They also discuss grounds for application of specific diuretic treatments in this syndrome. The review markedly helps to  systemize knowledge in this field.

I have the following critical comments:

-The authors describe well known processes, which are described in the textbooks while the new findings are not satisfactorily supported by references. More references supporting directly the statements presented in the paragraphs 2.1, 4.7, 4.7.1, 4.7.2 should be introduced in the text. 

-The authors should carefully check the citations. For instance in the reference 24 (Shapiro et al) the authors claim that aldosterone significantly contributes to sodium retention in nephrotic syndrome.

Author Response

(The authors gave the same response as above.)

Reviewer 3 Report

Comments and Suggestions for Authors

This is a comprehensive discussion from the clinical perspective for the treatment of nephrotic syndrome. There are some mixed discussions that can be separated for a clearer discussion, for example, in the Abstract, edema and hypervolemia are related but separate conditions, and the suggestion for the use of ENaC channel blocker can address the former and not exactly the latter.

Some discussions are not incorrect but somewhat digressive, for example, in lines 334-339, causes of diuretic resistance can be described with more focus on each physiological and mechanistic step, and not about pharmacokinetics or pharmacodynamics, which are a bit general.

There are also some technical details that can be revised, as specified below:

1. In Figure 1 and ll. 369-370: NKCC2 is the proper acronym and not NCCK2.

2. In Figure 1: Collecting duct would be more proper than collecting tubule. 

3. l. 383: It is unclear if metolazone is scarcely available indeed.

4. ll. 596-597: "(NaCl)." is not an acronym of NCC knockout mice and it should be deleted.

5. l. 627: re-sis- -> resis-

6. l. 630: ex-plora- -> explora-

Author Response

We are deeply appreciative of your expertise and guidance, which have significantly enriched the content and precision of our manuscript. I am writing to address the modifications suggested in your recent evaluation of our manuscript. Each modification has been addressed and is visually demarcated by red markings for enhanced clarity.

- We have revised the Abstract to incorporate the necessary textual modifications, as indicated in Section I. 25-27

- Your suggestions regarding the section detailing the causes of diuretic resistance have been diligently implemented. The requisite modifications have been made in paragraph 3., Section 323-339

- Furthermore, all technical details highlighted by you have been thoroughly addressed as outlined below:   

    - NKCC2 and NKCC1 changed to NCCK2 and NCCK1

    - collecting tubule changed to collecting duct

    - “(NaCl)” is deleted with strikethrough

    - “However, metolazone is scarcely available” is deleted with strikethrough

    - I. 655  re-sis changed to resis

    - I. 658 ex-plo changed to explo

Thank you once again for your expertise and assistance throughout this process. We are grateful for the opportunity to refine our manuscript in accordance with your esteemed insights. Should you require any further clarification or have additional recommendations, please do not hesitate to reach out.

Round 2

Reviewer 1 Report

Comments and Suggestions for Authors

Thank you for updating the manuscript.